# Antiproliferative and Cytotoxic Properties of Propynoyl Betulin Derivatives against Human Ovarian Cancer Cells: In Vitro Studies

**DOI:** 10.3390/ijms242216487

**Published:** 2023-11-18

**Authors:** Ewa Chodurek, Arkadiusz Orchel, Paweł Gwiazdoń, Anna Kaps, Piotr Paduszyński, Marzena Jaworska-Kik, Elwira Chrobak, Ewa Bębenek, Stanisław Boryczka, Janusz Kasperczyk

**Affiliations:** 1Department of Biopharmacy, Faculty of Pharmaceutical Sciences in Sosnowiec, Medical University of Silesia in Katowice, 8 Jedności Str., 41-208 Sosnowiec, Poland; aorchel@sum.edu.pl (A.O.); pgwiazdon@wp.pl (P.G.); akaps@sum.edu.pl (A.K.); ppaduszynski@sum.edu.pl (P.P.); jkik@sum.edu.pl (M.J.-K.); janusz.kasperczyk@sum.edu.pl (J.K.); 2Department of Organic Chemistry, Faculty of Pharmaceutical Sciences in Sosnowiec, Medical University of Silesia in Katowice, 4 Jagiellońska Str., 41-200 Sosnowiec, Poland; echrobak@sum.edu.pl (E.C.); ebebenek@sum.edu.pl (E.B.); boryczka@sum.edu.pl (S.B.)

**Keywords:** ovarian cancer, betulin derivatives, cytotoxicity, regulated cell death

## Abstract

Due to the incidence of ovarian cancer (OC) and the limitations of available therapeutic strategies, it is necessary to search for novel therapeutic solutions. The aim of this study was to evaluate the cytotoxic effect of betulin **1** and its propynoyl derivatives **2–6** against ovarian cancer cells (SK-OV-3, OVCAR-3) and normal myofibroblasts (18Co). Paclitaxel was used as the reference compound. The propynoyl derivatives **2–6** exhibited stronger antiproliferative and cytotoxic activities compared to betulin **1**. In both ovarian cancer cell lines, the most potent compound was 28-propynoylbetulin **2**. In the case of compound **2**, the calculated IC_50_ values were 0.2 µM for the SK-OV-3 cells and 0.19 µM for the OVCAR-3 cells. Under the same culture conditions, the calculated IC_50_ values for compound **6** were 0.26 µM and 0.59 µM, respectively. It was observed that cells treated with compounds **2** and **6** caused a decrease in the potential of the mitochondrial membrane and a significant change in cell morphology. Betulin **1**, a diol from the group of pentacyclic triterpenes, has a confirmed wide spectrum of biological effects, including a significant anticancer effect. It is characterized by low bioavailability, which can be improved by introducing changes to its structure. The results showed that chemical modifications of betulin **1** only at position C-28 with the propynoyl group (compound **2**) and additionally at position C-3 with the phosphate group (compound **3**) or at C-29 with the phosphonate group (compound **6**) allowed us to obtain compounds with greater cytotoxic activity than their parent compounds, which could be used to develop novel therapeutic systems effective in the treatment of ovarian cancer.

## 1. Introduction

Ovarian cancer was the third most common cancer of the female reproductive system, with the age-standardized incidence rate estimated at 6.6 per 100,000 women in 2020 [1]. Moreover, according to estimates from the report of the World Health Organization on the basis of the GLOBOCAN database, 313,959 new ovarian cancer cases and 207,252 ovarian cancer deaths occurred in 2020 [2]. Systemic chemotherapy after primary debulking cytoreductive surgery is the most commonly used regimen in the treatment of OC. The development of drug resistance leads to a decrease in the intracellular concentration of drugs in cancer cells, and it affects their cell cycle, regulated cell death (RCD) ability, and signaling pathways [3]. The key element of effective antitumor therapy is the ability of the active substance to induce cell death in cancer cells [4]. In healthy tissues, homeostasis is maintained due to a balance between cell proliferation and cell death. Both of these processes are strictly regulated, and dysfunctions in their control mechanisms play a significant role in the pathogenesis of several disorders, including cancer [5]. Mammalian cells possess complex, genetically encoded, molecular mechanisms responsible for the elimination of redundant, superfluous, damaged, or infected cells. According to the recommendations of the Nomenclature Committee on Cell Death 2018, 12 different types of RCD have been identified based on the underlying molecular pathways [6]. They include classical intrinsic and extrinsic apoptosis, autophagy-dependent cell death, as well as processes collectively classified earlier as regulated necrosis (mitochondrial permeability transition (MPT)-driven necrosis, necroptosis, pyroptosis, etc.). Programmed cell death is defined as a kind of RCD occurring as a consequence of normal organ development or physiological cell turnover. On the other hand, RCD can be initiated by various disturbances in the extracellular or intracellular microenvironment if their magnitude overcomes cellular mechanisms enabling adaptation to stress conditions. Contrary to popular belief, neoplastic cells are not necessarily more resistant to apoptosis induction than normal cells. Actually, the efficacy of many anticancer therapies depends on their capacity for RCD activation in transformed cells [7]. Unfortunately, cancer progression to more advanced stages is often accompanied by the acquisition of some genetic or epigenetic alterations, resulting in a significantly diminished sensitivity of transformed cells to RCD-inducing stimuli. That phenomenon contributes to the emergence of treatment resistance and may be responsible for the failure of therapy [8]. Biochemical abnormalities underlying apoptosis resistance include: hindered death receptor expression or activity, increased expression antiapoptotic proteins (Bcl-2, Bcl-xL, Mcl-1, A1/Bf1, cFLIP, PED/PEA-15, and IAPs), impaired function of proapoptotic proteins (p53, Bax, BH3 only proteins, and Apaf-1), high chaperoning activity of hsp90, upregulation of proteasome activity, and overexpression of galectins, among others [8,9]. Altered protein functions may result from gene mutations or epigenetic gene regulations, such as CpG-island hypermethylation of gene promoters, deacetylation of histone proteins, or the action of miRNAs. Seborova et al. [10] pointed out the potential use of miRNA molecules and long non-coding RNAs in predicting the development of ovarian cancer and its metastasis. An increasing number of publications indicate that protein overexpression and changes in signaling pathways, including not only involved in an apoptosis progression but also in drug efflux or damage and DNA repair, may lead to the development of chemoresistant ovarian cancer [11]. However, restoring sensitivity to RCD in neoplastic cells emerges as an important cancer treatment strategy. For instance, interference in the B-cell lymphoma 2 (Bcl-2) protein network using Bcl-2 homology 3 (BH3) mimetics resulted in the development of new preparations, such as Venetoclax [12] and Navitoclax [13].

It is believed that various natural compounds, especially phytochemicals, may re-sensitize tumor cells to RCD due to a variety of unique mechanisms and pleiotropic effects on intracellular processes [8]. This category includes agents affecting mitochondrial function, such as betulin and its derivatives [14]. Betulin is a lupane-type triterpenoid with four six-membered rings and one five-membered ring and with three positions in the backbone, the C-3, C-19, and C-28, where chemical modifications can be easily performed [15,16]. It can be used as a starting material for the synthesis of successive derivatives, e.g., esters, phosphates, amides, carbamates, and sulfates, with tailored physicochemical and biological properties [17,18,19]. Some of them may exert remarkable antiviral, antibacterial, anti-inflammatory, antioxidant, hepatoprotective, and even anticancer properties useful for developing new therapeutic strategies for patients [20,21].

The aim of this study was to evaluate the in vitro anticancer activities of betulin and its propynoyl derivatives (28-O-propynoylbetulin, 3-O-diethoxyphosphorylbetulin-28-O-propynoylbetulin, 28-O-diethoxyphosphoryl-3-O-propynoylbetulin, 30-diethoxyphosphoryloxy-28-O-propynoylbetulin, and E-29-diethoxyphosphoryl-28-O-propynoylbetulin) against human ovarian cancer cell lines (SK-OV-3 and OVCAR-3). The search for new active compounds is crucial for the development of novel therapeutic strategies in the treatment of ovarian cancer.

## 2. Results and Discussion

Cancer is one of the leading causes of premature death in women. According to the GLOBOCAN factsheet, 9.2 million new cases and 4.4 million deaths from various types of cancer were reported in the woman population in 2020 worldwide [22,23]. Due to the non-specific symptoms of OC, such as a feeling of fullness and gastric problems, abdominal or pelvic pain, urinary tract ailments, and no effective screening strategies, it is usually diagnosed at an advanced stage [24,25]. Platinum- and taxane-based drugs are most commonly used in ovarian cancer chemotherapy. Due to either the formation of adducts of purines or the hyperstabilization of microtubules, they could initiate cell death and limit tumor progression. In the treatment of advanced forms of OC, the possibility of using paclitaxel with a humanized monoclonal antibody, bevacizumab, that targets the vescular endothelial growth factor and has an anti-angiogenic activity [26], and poly (ADP-ribose) polymerase (PARP) inhibitors capturing the PARP-1 protein, causing the accumulation of PARP-DNA nucleoprotein complexes and cancer cell death [27], should also be considered. Unfortunately, the use of chemotherapeutics is limited because of the high risk of incidence of adverse hematological effects, ototoxicity, nephrotoxicity, and neurotoxicity in patients [28]. Moreover, the probability of developing chemoresistance increases with successive cycles of therapy, which lowers the patient’s survival rate [28,29]. Understanding the mechanisms leading to the development of OC and finding a way to overcome its resistance to therapy are some of the key challenges of modern medicine.

Betulin **1** (lup-20(29)-ene-3β,28-diol, Figure 1) is a widely available triterpenoid with a number of biological properties, including proven anticancer activity. The cytotoxic and antiproliferative activity of betulin **1** has been studied in many established cancer cell lines (e.g., SK-N-AS, FTC 238, A549, MCF-7, HT-29, and A431) and primary tumor cultures (e.g., HPOC, HPCC, and HPGBM) [30]. If an insufficient antiproliferative activity is observed, the solution is to either introduce structural changes to betulin **1** that may improve its pharmacological activity or to combine therapy with ionizing radiation or known chemotherapeutics, such as cisplatin and doxorubicin [31]. 

In this study, all of the used betulin derivatives have a propynoyl group either at the C-28 (compounds **2**, **3**, **5**, and **6**) or at the C-3 (compound **4**), respectively (Figure 1). The effects of additional substitutions based on the introduction of diethoxyphosphoryl or diethoxyphosphoryloxy moieties on anticancer activity of compounds **3**–**6** were also evaluated. Phosphorus is a compound commonly found in the human body involved in various cellular processes, including membrane integrity, cell signaling, and cell metabolism [32]. The presence of phosphate or phosphonate groups may primarily affect the metabolic stability of compounds and their transport across plasma membranes. Due to the negative charge of these moieties, the temporary protecting groups are used in prodrug synthesis to increase their bioavailability, penetration, and specificity and to reduce the risk of their premature degradation in the patient’s body [33]. Inside the targeted cells, the protecting groups are rapidly degraded in an enzymatic and/or chemical deprotection manner [34]. Previous studies have shown that the introduction of a group with a phosphorus atom into a known drug molecule can improve the biological activity and even enable the acquisition of anticancer properties compared to the original structure [35]. The effect of the type and location of specific functional groups in the modified betulin **1** on anticancer activity in OC cells should be precisely investigated.

The inhibition of OC cell growth after the administration of betulin **1** and its propynoyl derivatives **2–6** was assessed with the use of sulforhodamine B (SRB) staining. The presented data (Figure 2 and Figure 3; Appendix A) clearly showed that 28-propynoylbetulin **2** had the strongest antiproliferative activity against both SK-OV-3 and OVCAR-3 cell lines.

OVCAR-3 cells seem to be more sensitive to compound **2** because the observed antiproliferative effect was stronger. In both cell lines, a decrease in the number of cells was evident from a concentration of 0.3 µM. The 28-propynoyl derivatives **3**, **5**, and **6** with an additional group containing a phosphorus atom showed a strong antiproliferative activity at concentration 1 µM, and it was higher against SK-OV-3 cells. Although derivative **4** (with a propynoyl group at the C-3 position) showed a slightly lower activity against SK-OV-3 cells, it was still stronger than unmodified betulin **1**. Almost identical results were observed for the OVCAR-3 line. The derivative **4** caused an inhibition of proliferation above a concentration of 3 µM. For comparison, betulin **1** had an effective antiproliferative effect at a concentration of 10 µM after a 5-day treatment period. Obtained data from the SRB assay after the 3- and 5-day incubation period were used to calculate IC_50_ values (concentration of the compound required for 50% of its maximal inhibitory effect), which are presented in Table 1.

Generally, the estimated IC_50_ values were lower in SK-OV-3 than in OVCAR-3 after treatment with betulin **1** and compounds **3**, **4**, **5**, and **6**. The compounds form the following rank order of antiproliferative activity against SK-OV-3 and OVCAR-3 cells after 5 days of treatment: **2** > **6** > **5** > **3** > **4** > **1** and **2** > **3** > **6** > **5** > **4** > **1**, respectively. The results of numerous studies have confirmed that unmodified betulin **1** may be an inhibitor of cancer cell growth. However, not all tumor cell lines are betulin sensitive; hence, it has been suggested that the introduction of some functional groups may increase the chances of successful anticancer therapy [18,30,36]. 

The obtained results indicate a significantly greater usefulness of betulin derivatives with a propynoyl group at the C-28 position compared to the C-3 position in antiproliferative therapy. In previous studies, Kaps et al. [37] showed a strong inhibition of human melanoma cell lines C32 and A2058 proliferation by compound **2** at a concentration of 3 μM. Orchel et al. [38] demonstrated that compounds **2** and **5** had a strong antiproliferative activity against SK-BR-3 and MCF7 breast cancer cell lines, achieving the lowest IC_50_ values of 2.12 µM and 5.34 µM, respectively, after a 3-day incubation period. In turn, Chrobak et al. [35] showed the high anticancer activity of compound **6** against T47D, C32, and SNB-19 cell lines, and the obtained IC_50_ values were 0.7 μM, 0.6 μM, and 0.43 μM, respectively. Significant tumor growth inhibition with compound **3** was reported in A549, DU145, Hs294T, MV-4-11, CCRF/CEM, and P388 cells [39]. 

After confirming the antiproliferative activity of the compounds in the SK-OV-3 and OVCAR-3 cell lines, our efforts focused on the assessment of induction of RCD by the tested ones. During the early stages of RCD, MPT pore opening, outer membrane permeabilization, release of cytochrome c, and activation of effector caspases may occur. Depending on the activity of the inducible NF-κB (nuclear factor kappa-light-chain-enhancer of activated B cells) transcription factor, the expression of members of the Bcl-2 and IAP protein families, as well as the intracellular concentration of Ca^2+^ and ATP (adenosine triphosphate), switching between apoptosis and MPT-mediated necrosis may occur in cells. For example, completion of the apoptosis program requires maintaining a certain ATP pool involved in, among other things, caspase activation. Complete depletion of ATP is associated with the development of necrotic-like cell response [40,41,42]. OC cells, like many other neoplastic cells, have a certain reduced sensitivity to apoptosis compared to normal cells. The results of previous studies showed a high level of the antiapoptotic protein Bcl-2 and a low level of caspases-3, -8, and -9 associated with the initiation and execution of apoptosis in OC cells. It was also observed that downregulation of a member of the IAP family, X-linked inhibitor of apoptosis (XIAP), may induce p53-mediated apoptosis and sensitize OC cells to the action of cisplatin. It was manifested by an increase in the population of apoptotic cells in the tested material. Silencing XIAP with antisense oligonucleotides led to an increase in caspase-3 activity and an increase in PARP cleavage, and it could enhance the proapoptotic effect in cancer cells. Other factors that may regulate apoptosis in OC cells include survivin that inhibits caspase-3 and -7 and the Fas-related interleukin-1β-converting enzyme (FLICE) inhibitory protein (FLIP) that can prevent the caspase-8 and -3 activation. Overexpression of these factors may reduce the sensitivity of OC cells to taxanes and to platinum derivatives [9]. Anaya-Eugenio et al. [43] reported that betulin **1** induced a loss of mitochondrial membrane potential and increased the level of proapoptotic members of the Bcl-2 family in various cancer cell lines. It was also capable of sequentially activating caspase-9, -3, and -7, and cleaving PARP. This may be the result of the upregulation of factors belonging to the p53 family in cancer cells. The ability to induce apoptosis by betulin **1** has been demonstrated, inter alia, in A549, B164A5, Jurkat, and HeLA cancer cell lines.

This study estimated the death of OC cells based on the integrity of cell membranes, DNA fragmentation, and cell morphology after incubation with compounds with expected cytotoxic activity. The release of lactate dehydrogenase (LDH, a cytosolic enzyme) into the culture medium is an indicator of advanced changes in the integrity of cell membranes, which characterize dying cells. As shown in Figure 4, in SK-OV-3 cells, significant amounts of LDH in culture medium and, thus, a strong cytotoxic effect of compounds **2**, **3,** and **5** at all concentrations were observed. It is worth noting that the most cytotoxic agent used in SK-OV-3 cells was compound **6** at both 10 μM and 30 μM.

In the OVCAR-3 cell line, the strongest cytotoxicity was observed in cells treated with compound **3**. Compounds **2**, **5,** and **6** also showed significant cytotoxic activity against OVCAR-3 cells at all concentrations. The loss of cell membrane integrity was also confirmed for compound **4** at a concentration of 10 μM and 30 μM and for betulin **1** at a concentration of 30 µM. The introduction of an alkyne moiety with triple bond into the structure of betulin **1** increased the cytotoxicity of compounds **2**–**6** in both cell lines. The presence of an additional diethoxyphosphoryl or diethoxyphosphoryloxy moiety slightly increased the cytotoxic effect. In conclusion, the compound with the weakest cytotoxic activity was compound **4**, in which the propynoyl group was introduced at the C-3 position and the diethoxyphosphoryl group at the C-28 position. Similarly to the SRB results, the response of OVCAR-3 cells to the test substances was much weaker, and the amounts of released enzyme were lower than in SK-OV-3. It suggests that each type of cell line treated with the same compounds may differ in protein expression and cell death pattern. Only betulin **1** showed a high similarity of the percentage of cytotoxicity for both cell lines at all concentrations. The results obtained in the LDH assay correlate with the SRB staining results for SK-OV-3 and OVCAR-3 cells. 

To determine the relation between a decline in cell viability and the activation of RCD pathways, the effect of betulin **1** and compounds **2**–**6** on DNA fragmentation was assessed. It is assumed that in RCD cells, internucleosomal DNA is cleaved by caspase-activated DNase (CAD), and it can be detected in cell lysates as mono- and oligonucleosomes. The assay is based on the assumption that the cell membrane remains tight for a relatively long time during the successive stages of apoptosis, and cytoplasmic components do not leak out of the cell, so that the investigated histone–DNA complexes can be determined [44,45]. The graph (Figure 5) shows the ratio of the number of mono- and oligonucleosomes, referred to as the enrichment factor (EF), in the samples. The higher the EF value, the more intense the cell death in the analysed material. 

As shown in Figure 5, betulin **1** caused a significant increase in the amount of internucleosomal DNA fragments in SK-OV-3 cells. Moreover, the observed effect seems to be much stronger at 10 µM than at 30 µM. The apparent drop in the EF index at 30 µM is likely due to the potent cytotoxic effect of betulin **1** at this concentration. This probably resulted in an increase in the permeability of cell membranes and the loss of a part of the study material along with the removed culture medium. Among the derivatives, moderate increases were noticed after 24 h of exposure of the cells to compounds **2** and **6**, and they were concentration dependent. OVCAR-3 cells, compared to SK-OV-3 cells, seem to be less sensitive to tested compounds, and they were characterized by significantly low values of the EF. In the case of betulin **1**, only the highest concentration (30 µM) caused an increase in DNA fragmentation, while the lower concentrations, both 3 µM and 10 µM, showed a number of mono- and oligonucleosomes comparable to the control. This indicates that the sensitivity to the betulin **1** is dependent on the cell type. Among the betulin derivatives, compound **4** with a diethoxyphosphoryl group at the C-28 position had the greatest effect on the degradation of genetic material, and the effect was concentration dependent. Compound **6** has a slightly smaller effect than the lowest concentration of compound **4**, and the result was quite similar for both 3 µM and 10 µM. Knowing that propynoyl derivatives **2**–**6** at concentrations above 3 µM promoted cell death more strongly than betulin **1**, and the EF values in cells treated with derivatives were lower than in cultures containing betulin **1**, it can be assumed that apoptosis is not the only type of RCD death in OC cells. It is worth mentioning that in the late stages of apoptosis, the cell membrane is disrupted and the cell integrity is lost (referred to as secondary necrosis) [46], which may lead to the loss of some of the complexes in the obtained material and lower the EF values in the test samples. Therefore, the results of DNA fragmentation should be confronted with the morphological changes of transformed cells after incubation with test compounds. The characteristic morphology of apoptotic cells can be observed under the microscope after staining with a fluorescent DNA intercalating dye, e.g., acridine orange. To study this issue, SK-OV-3 cells were used, and compound **4**, as it was apparently less toxic, was omitted in assays. Figure 6 shows the cells treated for 24 h with tested compounds and subsequently stained with acridine orange. 

Control cells (Figure 6a) contained nuclei with a loose chromatin network typical for healthy interphase cells. Treatment with betulin **1** resulted in an emergence of cells displaying typical apoptotic morphology: chromatin condensation (pyknosis) and fragmentation (karyorrhexis), as well as shrinkage of the cytoplasm [47]. Surprisingly, cells with clearly apoptotic morphology were sparse in cultures treated with betulin derivatives (Figure 6c–f). Cells were often swollen, and their nuclei presented two types of morphological changes. Some cells contained clearly pyknotic (condensed) but not fragmented nuclei (anucleolytic pyknosis) [48]. In some cells, nuclei were distended and weakly stained, suggesting karyolysis. These observations indicate that in this case, cell death possibly resulted from the process of necrosis. According to current views, necrosis may represent a regulated cellular response to stress both in physiological and pathological conditions [49]. Certain processes traditionally considered as apoptosis hallmarks, such as internucleosomal DNA fragmentation [50,51] or activation of caspases [52], were sometimes found in necrotic cells. The same agents can induce both apoptotic and necrotic cell demise depending on their concentration or cell energy status. Such observations gave rise to the concept of an apoptosis–necrosis continuum, a biochemical network shared by these modes of cell death [53]. Possibly, there exists a spectrum of RCD programmes, and typical apoptosis and necrosis represent its extremes. The observed cytomorphological features (like cell rounding, shrinkage, and nucleolytic pyknosis) as well as intense DNA fragmentation after betulin **1** treatment were characteristic for apoptotic cells.

Mitochondria play a crucial role in switching on several RCD forms, especially intrinsic apoptosis and MPT-driven necrosis [6]. Complete execution of the apoptotic program requires providing some amount of ATP [54]. Drugs capable of inducing a rapid opening of MPT pores can destroy mitochondria and interrupt the synthesis of ATP. This may be the first step in the process leading to necrotic cell death [55]. It has been shown that betulin and betulinic acid activate the process of intrinsic apoptosis associated with cytochrome c and Smac protein release, reactive oxygen species generation, as well as the activation of caspase-9 and -3 [56,57,58,59,60]. However, there are some controversies related to the mechanisms underlying that phenomenon. Some studies have shown that apoptosis of cells treated with the abovementioned substances was accompanied by a rapid dissipation of mitochondrial membrane potential and seemed to depend on the function of MPT pores [56,58]. Although MPT pores’ opening is rather linked with MPT-driven necrosis, it can also result in cell apoptosis (under submaximal MPT, if some energy supply is retained) [55]. Some reports suggest that betulinic acid (but not betulin) directly influences mitochondria and triggers the permeability transition. However, Potze et al. [22] revealed the ability of betulinic acid to inhibit stearoyl-CoA-desaturase (SCD-1), an enzyme located in the endoplasmic reticulum. As a consequence, the degree of saturation of cardiolipin (the principal phospholipid of the mitochondrial membrane) increased and mitochondria underwent ultrastructural changes and released cytochrome c. Ultimately, cell death was induced.

To test if the appearance of manifestations of cell death (in SK-OV-3 cell cultures treated with the most cytotoxic betulin derivatives) was preceded by disturbances in mitochondrial function, we used tetramethylrhodamine methyl ester (TMRM) vital imaging to estimate mitochondrial membrane potential. Additionally, we tested an effect of the blockage of MPT pore opening with CsA on cytotoxic activity of the compound **6** [59]. As shown in Figure 7, the incubation of cells with betulin derivatives for 3 h resulted in the broad disappearance of red fluorescence, proving the dissipation of the mitochondrial inner membrane potential. 

Treatment of the SK-OV-3 cells with CsA almost completely prevented the derivative-**6**-induced cell death, which was evaluated on the basis of DNA fragmentation (Figure 8a) and plasma membrane permeability (measurement of DNA–histone complexes both in cytoplasm and in cell culture medium; Figure 8b).

This suggests that betulin derivatives caused a rapid disturbance in mitochondrial function resulting in deep energy deficit of cells and, eventually, their necrotic-like death. The results indicated the dominant role of necrosis over apoptosis in the case of betulin derivatives (compounds **2**, **3**, **5,** and **6**). For many years, apoptosis was considered the most favorable type of cell death due to the lack of the observed pro-inflammatory changes. Macrophages ingesting apoptotic cell remnants are able to present processed antigens to lymphocytes, but lack of a concomitant pro-inflammatory environment averts their activation. It prevents development of autoimmunity in the case of cell death resulting from normal cell turnover. However, within an inflammatory environment, there are more favorable conditions to initiate antitumor immune response. The balance between necrosis and apoptosis decides if the area of cell death becomes pro-inflammatory or anti-inflammatory. Cell necrosis is associated with large-scale release of intracellular contents containing some signals recognized by immune cells, such as Hsp70 protein, calreticuline, or oligonucleosomes. This may lead to recruitment of a more appropriate set of immune cells and breaking the tolerance to tumor-associated antigens [61]. Due to the frequently developing resistance of cancer cells to chemotherapy, not only alternative compounds are sought, but also RCD mechanisms that stimulate and multiply the patient’s antitumor immunogenicity.

## 3. Materials and Methods

### 3.1. Reagents

Betulin **1** (Figure 1) with purity ≥ 98% was purchased from Sigma Aldrich (Poznań, Poland). Betulin derivatives **2**–**6** (Figure 1) were obtained from the Department of Organic Chemistry, Faculty of Pharmaceutical Sciences in Sosnowiec, Medical University of Silesia in Katowice. The detailed synthesis was described in the earlier literature [15,41,42]. Derivatives **2**–**6** were purified through column chromatography using silica gel and mixtures of dichloromethane or chloroform and ethanol in appropriate proportions as eluent. The purity of the target compounds was assessed by performing ^1^H, ^13^C, and ^31^P NMR spectra, IR spectra, and MS analysis [18,38,39].

Paclitaxel (Sigma Aldrich), at concentrations of 0.1–100 µM, was used as a reference compound in part of the studies. 

### 3.2. Cell Cultures

This study used ovarian cancer cell lines, which are part of the panel of cell lines used as part of the NCI-60 Human Cell Lines Screen program, characterized by overexpression of the c-erbB2/neu (HER2) receptor [62,63]. The human ovarian cancer cell lines (SK-OV-3 and OVCAR-3) were purchased from the American Type Culture Collection (ATCC^®^, Manassas, Virginia, USA). SK-OV-3 cells (ATCC No. HTB-77™) were cultured in McCoy’s 5A medium containing D-glucose (3 g/L), L-glutamine (0.219 g/L), and sodium bicarbonate (2.2 g/L) supplemented with 10% fetal bovine serum, 10 mM HEPES (pH 7.3–7.5), 100 U/mL penicillin, and 100 μg/mL streptomycin (all from Sigma Aldrich). 

OVCAR-3 cells (ATCC No. HTB-161™) were cultured in RPMI-1640 medium containing HEPES buffer (2.383 g/L), sodium bicarbonate (1.5 g/L), sodium pyruvate (0.11 g/L) supplemented with 20% fetal bovine serum, 10 µg/mL bovine insulin, 100 U/mL penicillin, and 100 μg/mL streptomycin (all from Sigma Aldrich). 

Next, 18Co cells (ATCC No. CRL-1459™) were cultured in MEM medium containing HEPES buffer, 10% fetal bovine serum, 100 U/mL penicillin, and 100 μg/mL streptomycin (all from Sigma Aldrich).

Cultures were maintained at 37 °C in a humidified atmosphere containing 5% CO_2_.

### 3.3. Cell Proliferation—SRB Assay

Betulin **1** and its derivatives were dissolved in dimethyl sulfoxide (DMSO; Sigma Aldrich) as a 15 mM stock solution. Working solutions were prepared through the dilution of stock solutions with culture medium. The final DMSO concentration in culture media was adjusted to 0.2%. The effect of tested compounds on the proliferation of SK-OV-3 and OVCAR-3 cells was measured using the In Vitro Toxicology Assay Kit, Sulforhodamine B based (Sigma Aldrich). Cells were seeded in 96-well plates (Greiner Bio-One, Kremsmünster, Austria) at an initial density of 3 × 10^3^ cells per well in 200 µL of culture medium. After 24 h of incubation, betulin **1** and its derivatives at a concentration range of 0.1 μM to 30 μM were added to the wells. Control cultures were treated with vehicle alone (0.2% DMSO). The incubation periods for cells with test compounds were 1 day, 3 days, and 5 days. After appropriate incubation periods, culture media were removed, and cells were fixed in 10% trichloroacetic acid solution for 60 min at 4 °C. After fixation, the TCA solution was removed, and the plates were washed three times with deionized water (300 μL/well) and dried. At the next step, cells were stained with 0.4% sulforhodamine B for 30 min. The excess dye was removed by washing three times with 1% (vol/vol) acetic acid. The protein-bound dye was dissolved in 200 μL 10 mM Tris base solution. Absorbance was measured at 570 nm and 690 nm (reference wavelength) using the MRX Revelation plate reader (Dynex Technologies, Chantilly, VA, USA).

### 3.4. LDH Release Assay

In order to evaluate the cytotoxic effect, the release of LDH into the culture medium using the In Vitro Toxicology Assay Kit, Lactic Dehydrogenase Based (TOX7; Sigma Aldrich) was assessed. SK-OV-3 and OVCAR-3 cells were plated and cultured in 96-well plates (Greiner Bio-One) at the initial density of 5 × 10^3^ cells per well in 100 µL of culture medium. The density of the cells was determined in the same manner as in the SRB assay. After 24 h, the media were replaced with the working solutions, which contained appropriate test compounds at concentrations of 3 μM, 10 μM, 30 μM, and control medium (0.2% DMSO). Cells were exposed to the test substances for 24 h. The background control comprised wells without cells with only control medium. To determine the maximal LDH release (high control), half of the control cultures were treated with lysis buffer. Wells with untreated cells were used as the low control of the assay. LDH activity in media was assessed according to the manufacturer’s protocol. The absorbance was measured at wavelengths of 490 nm and 690 nm (background), and the percentage of released enzyme was calculated for cells growing in the presence of compounds according to the formula:Cytotoxicity [%] = [(A_T_ − A_L_)/(A_H_ − A_L_)] × 100
where A_T_ is the absorbance of treated wells; A_L_ is low control; and A_H_ is high control (maximal LDH release).

Cytotoxicity was expressed as a percentage of the released LDH calculated according to the difference between the absorbance of the treated and low control wells divided by the difference between the absorbance of the high control and low control wells. 

### 3.5. DNA Fragmentation Assay

Fragmentation of genomic DNA was determined by means of the Cell Death Detection ELISAPLUS kit (Roche, Basel, Switzerland). This method rests on the immunoenzymatic detection of DNA–histone complexes in the cytoplasmic cell fraction or cell culture supernatants. Cells were plated and cultured in 96-well plates (Greiner Bio-One) at the initial density of 5 × 10^3^ cells per well in 100 µL of culture medium. Cells were allowed to attach and grow for 1 day prior to the addition of test reagents. Then, the media were replaced with the working solutions (3 µM, 10 µM, 30 µM), and cells were exposed to compounds **2**–**6** for 24 h. In some cases, 5 µM cyclosporin A (CsA, an inhibitor of the mitochondrial permeability transition) was added to the cells 30 min before treatment with the betulin derivative. Subsequently, cells were incubated concomitantly with both compounds. After a 24 h incubation period, plates were centrifuged (200× *g*; 10 min), media were removed, and cells were lysed for 30 min at room temperature. At the next step, the lysates were again spun (200× *g*; 10 min), and 20 µL of supernatants (cytoplasmic fractions) was moved to the new 96-well streptavidin-coated plate. In one experiment, the presence of mono- and oligonucleosomes was determined in 20 µL samples of centrifuged culture supernatants (media) to detect the permeability of plasma membrane (considered as an indicator of cell necrosis). Then, the sandwich ELISA assay was fulfilled according to the manufacturer’s protocol. Absorbance was measured at 405 nm and 490 nm (reference wavelength), and the enrichment factor of the cytoplasmic (or supernatant) fraction in mono- and oligonucleosomes was calculated according to the formula: EF = absorbance of sample cells/absorbance of control cells.

A high EF index in the cytoplasmic fraction (resulting from DNA fragmentation while maintaining the tightness of cell membranes) is a typical indicator of the relatively early stage of apoptosis. 

### 3.6. Assessment of Cell Morphology

#### 3.6.1. Acridine Orange Staining

Acridine orange staining was preceded by preparation of a cell culture at a density 1 × 10^5^ per well in a 24-well glass bottom plate (NEST Biotechnology, Wuxi, China). Cells were cultured for 24 h in a standard medium (1 mL), after which the medium was replaced with a new medium containing 10 µM and 30 µM of the test compounds or with 0.2% DMSO (control). Cells were incubated in the presence of test substances for 24 h in a CO_2_ incubator. To each of the test wells, acridine orange dissolved in PBS (the final concentration was 1 µg/mL) was added and incubated for 5 min in a CO_2_ incubator. After incubation, the cells were observed under a Nikon Eclipse TS-100F microscope (Nikon Instruments Europe, Amstelveen, The Netherlands). The photographs were taken with a Nikon DS-Fi1 digital camera, supported by the NIS-Elements application.

#### 3.6.2. TMRM Analysis

Tetramethylrhodamine methyl ester (Sigma Aldrich) was used for the assessment of changes in the transmembrane potential of mitochondria. TMRM is a fluorescent cationic dye accumulating in mitochondria in a potential-dependent manner. Cell cultures were prepared as described above for acridine orange staining. Cells were exposed to test solutions for 3 h, and then TMRM was added (final concentration 100 nM) followed by incubation for 30 min in the CO_2_ incubator. Prior to observation, the plates were centrifuged (250× *g*, 5 min), and the medium was changed to PBS. The cells were analyzed under the Nikon Eclipse TS-100F microscope and photographed.

### 3.7. Statistical Analysis and Graphs

GraphPad Prism 10.0.2 (GraphPad Software Inc., San Diego, CA, USA) software was used for calculations and the statistical analysis of the IC_50_ values (Table 1). IC_50_ was determined by using the “nonlinear regression” mode, and the “log(inhibitor) vs. normalised response—Variable slope” equation was chosen in GraphPad Prism software for calculations. The obtained data were normalized to run from 100% down to 0%, and logarithms of concentration were used. Graphs (Figure 2, Figure 3, Figure 4, Figure 5 and Figure 8, and Appendix A) were also prepared using this software. In the case of the lack of normal distribution or homogeneity of variances, the remaining results were analyzed using a non-parametric Kruskal–Wallis test with multiple comparisons. Post hoc analysis was performed with the non-parametric equivalent of Tukey’s test. A *p* value of <0.05 was considered statistically significant. Otherwise, a one-way ANOVA followed by a Tukey’s post hoc test was applied. *t*-tests was used to perform pairwise comparisons between group means. Analysis was performed using Statistica 13.1 software (StatSoft, Tulsa, OK, USA). 

## 4. Conclusions

The presented results indicate effective anticancer activity of betulin derivatives against human ovarian cancer cell lines (SK-OV-3 and OVCAR-3). They confirm that chemical modifications of betulin **1** in the C-28 and C-3 positions with propynoyl and in the C-3, C-28, and C-30 positions with phosphate or C-29 phosphonate groups allow for obtaining compounds with greater cytotoxic activity than the parent compound, betulin **1**. The greatest effect was observed in the case of compounds **2** and **6**, which was most likely due to the presence of additional groups, propynoyl and/or diethoxyphosphoryl, in the betulin **1** structure. Compound **2**, containing a free hydroxyl group at the C-3 position and a propynoyl group at the C-28 position, showed high anticancer activity against both ovarian cancer cell lines. Moreover, it was shown that compound **6**, having an additional diethoxyphosphoryl moiety at the C-29 position, had significant activity towards SK-OV-3 cells. Compounds **2** and **6** had antiproliferative and cytotoxic activities, reduced the mitochondrial membrane potential, and significantly changed the cell morphology. Our results suggest that derivatives **2** and **6** can be used to develop new pharmaceutical dosage forms in the treatment of ovarian cancer.

## Figures and Tables

**Figure 1 ijms-24-16487-f001:**
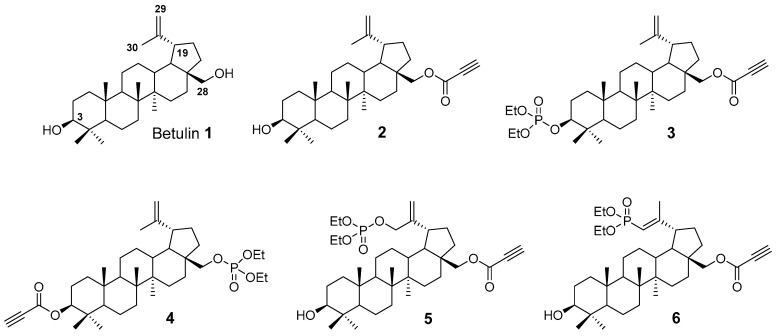
Chemical structure of betulin **1** and its propynoyl derivatives **2–6**.

**Figure 2 ijms-24-16487-f002:**
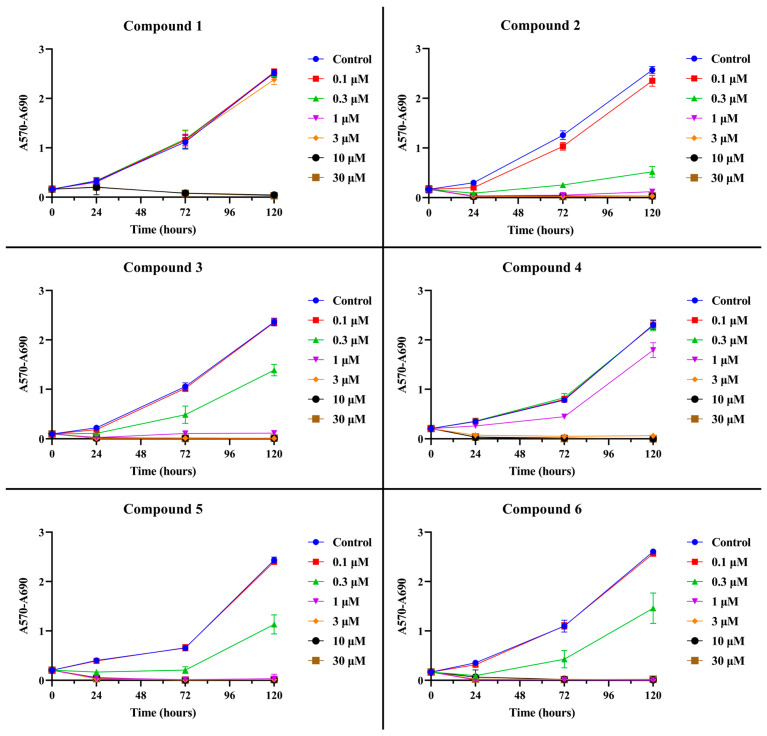
SK-OV-3 cell proliferation curves after exposition to betulin **1** and compounds **2**–**6** assessed using SRB assay. The series of curves represent SK-OV-3 cells’ proliferation measured in given time points: 1 day, 3 days, and 5 days of treatment with individual compounds at concentration range 0.1–30 µM. The plots are presented as the mean ± standard deviation.

**Figure 3 ijms-24-16487-f003:**
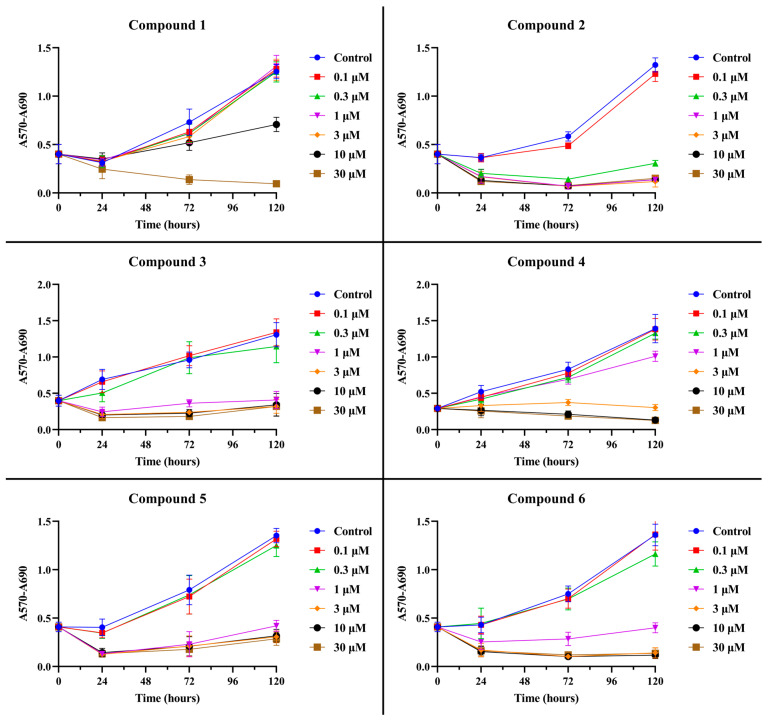
OVCAR-3 cell proliferation curves after exposition to betulin **1** and compounds **2**–**6** assessed using SRB assay. The series of curves represent OVCAR-3 cells’ proliferation measured in given time points: 1 day, 3 days, and 5 days of treatment with individual compounds at concentration range 0.1–30 µM. The plots are presented as the mean ± standard deviation.

**Figure 4 ijms-24-16487-f004:**
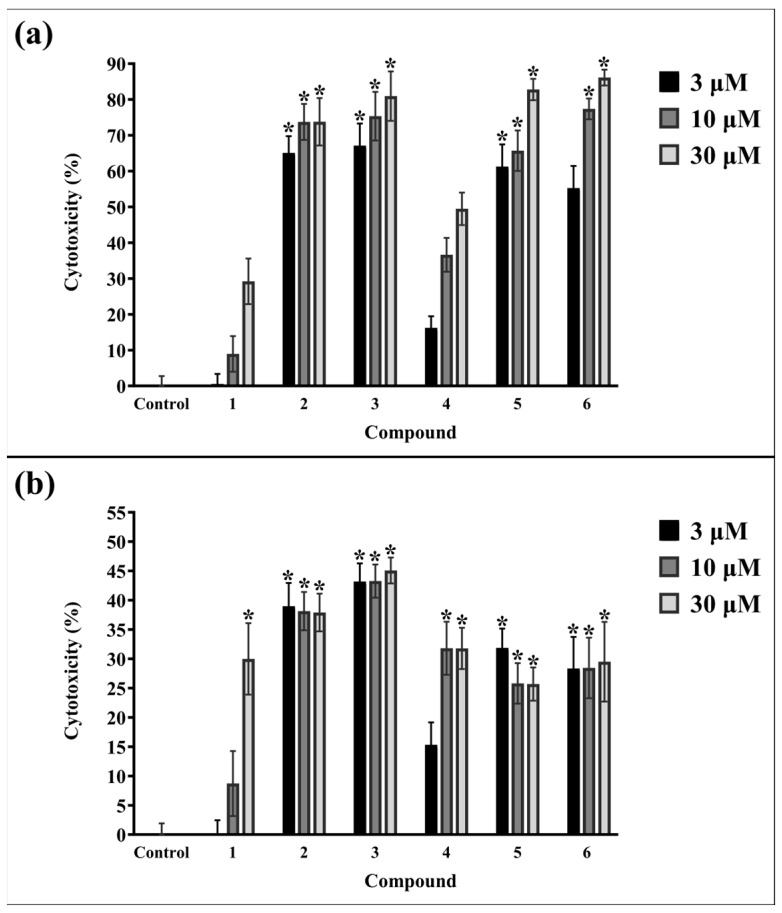
Cytotoxicity expressed as percentage of released LDH in (**a**) SK-OV-3 and (**b**) OVCAR-3 cells after 24 h incubation with betulin **1** and compounds **2–6**. The results are presented as the means ± standard deviation. * *p* < 0.05 vs. control.

**Figure 5 ijms-24-16487-f005:**
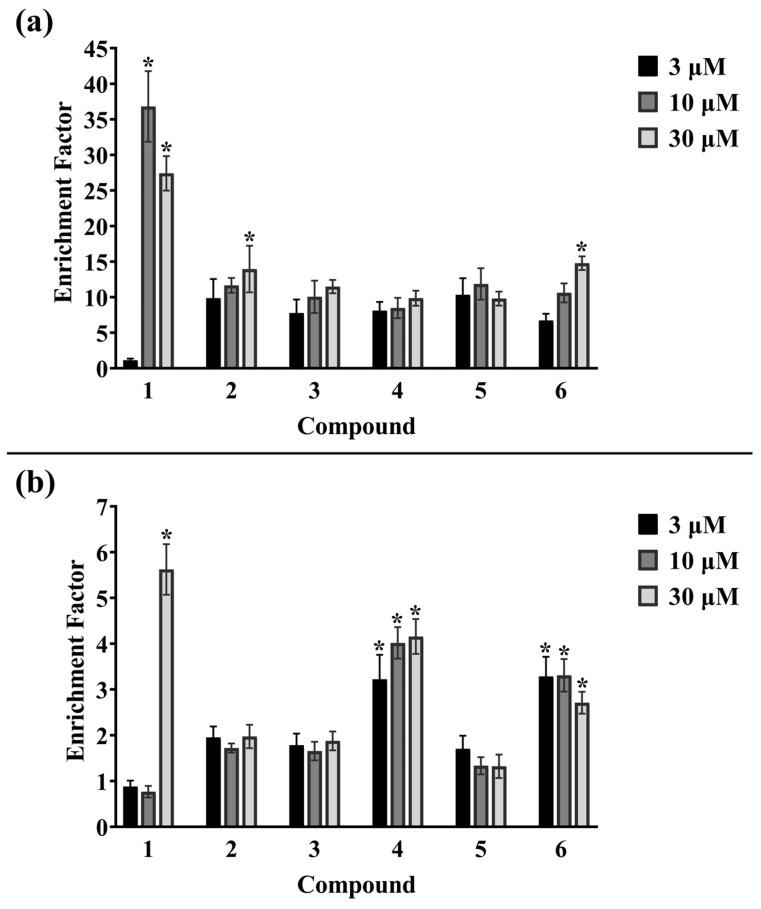
Determination of DNA fragmentation in (**a**) SK-OV-3 and (**b**) OVCAR-3 cells after 24 h incubation with betulin **1** and compounds **2–6**. The results are expressed as fold change over control and presented as the means ± standard deviation; * *p* < 0.05 vs. control.

**Figure 6 ijms-24-16487-f006:**
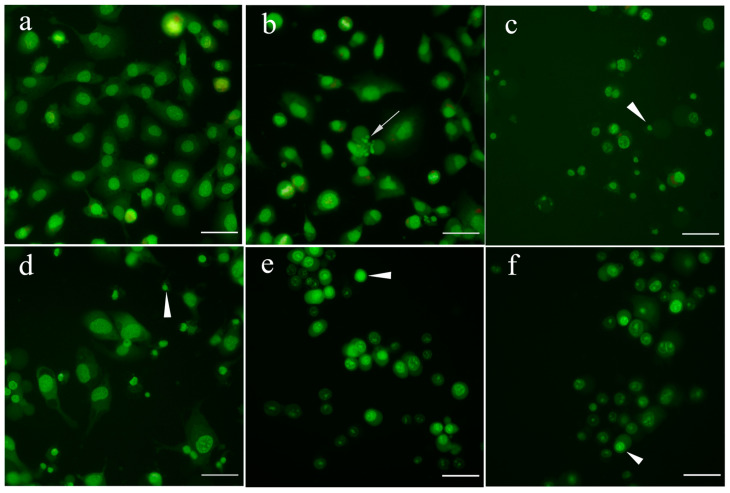
Effect of 24 h treatment with 30 µM betulins on morphology of SK-OV-3 cells. Cells were stained with acridine orange: (**a**) control cells; (**b**) betulin **1**; (**c**) compound **2**; (**d**) compound **3**; (**e**) compound **5**; (**f**) compound **6**. White arrows indicate the apoptotic cells; arrowheads indicate pyknotic cell nuclei (magnification 100×). Scale bars equal 100 μm.

**Figure 7 ijms-24-16487-f007:**
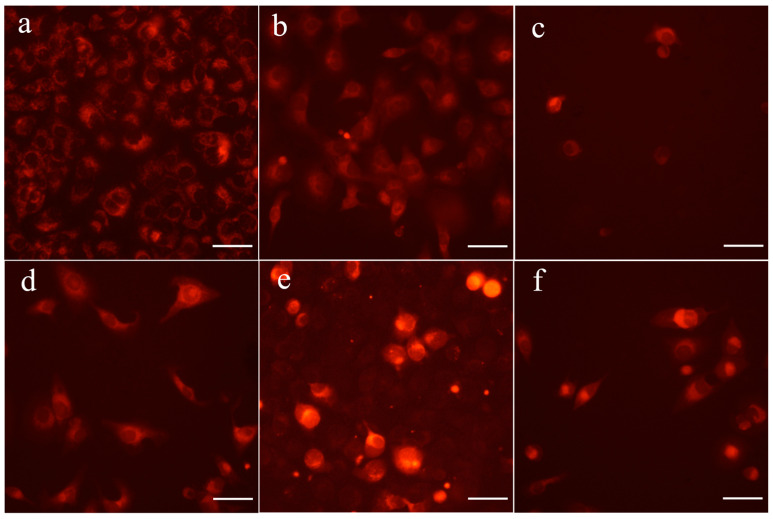
Effect of 3 h treatment with 30 µM betulins on the mitochondrial membrane potential in SK-OV-3 cells. Cells were stained with TMRM: (**a**) control cells; (**b**) betulin **1**; (**c**) compound **2**; (**d**) compound **3**; (**e**) compound **5**; (**f**) compound **6** (magnification 100×). Scale bars equal 100 μm.

**Figure 8 ijms-24-16487-f008:**
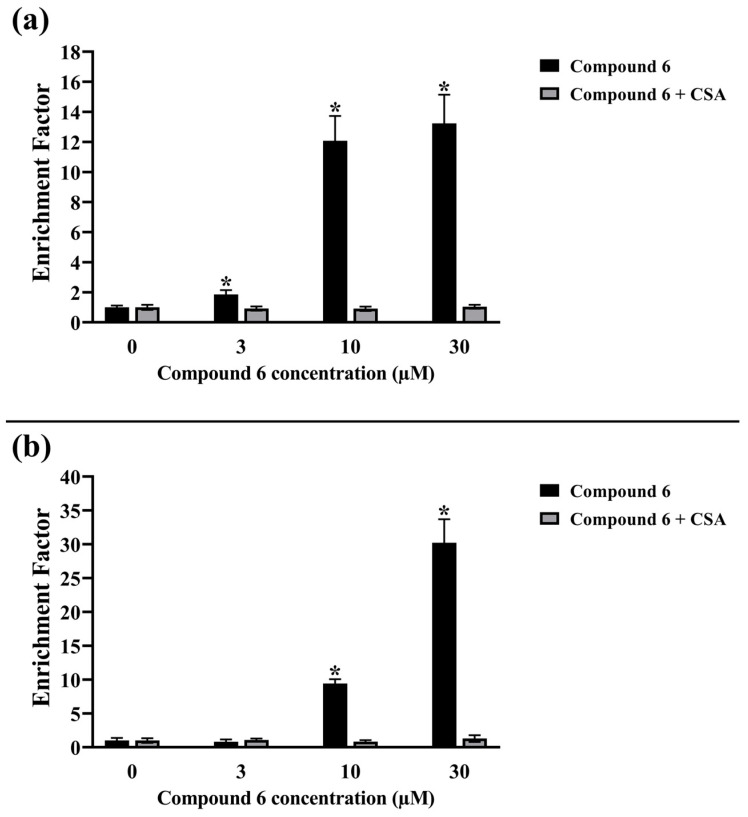
Evaluation of (**a**) the DNA fragmentation and (**b**) the plasma membrane permeability in SK-OV-3 cells incubated with compound 6 for 24 h alone or with CsA. The results are expressed as fold change over control and presented as the means ± standard deviation; * *p* < 0.05 vs. control. Lines show significant differences between untreated and CsA-treated cells.

**Table 1 ijms-24-16487-t001:** The in vitro antiproliferative activity of betulin **1**, compounds **2–6**, and paclitaxel expressed as the concentration of the compound required for 50% inhibition (IC_50_) on SK-OV-3 and OVCAR-3 cell lines and 18Co normal myofibroblasts after treatment for 3 days and 5 days, respectively. Neg: negative in the concentration used.

		Compound						
		1	2	3	4	5	6	PTX
Cell Line	Time	IC_50_ (μM)						IC_50_ (nM)
SK-OV-3	3 days	8.21	0.18	0.29	1.08	0.28	0.29	4.39
	5 days	4.59	0.20	0.34	1.33	0.29	0.26	5.65
OVCAR-3	3 days	9.03	0.16	0.90	1.88	0.54	0.68	4.49
	5 days	10.11	0.19	0.49	1.45	0.60	0.59	2.97
18Co	3 days	Neg	Neg	12.42	9.47	Neg	Neg	Neg
	5 days	Neg	Neg	Neg	23.72	Neg	Neg	14.79

## Data Availability

The data presented in this study are available upon request from the corresponding author.

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
