# Peer review of "Antiproliferative and Cytotoxic Properties of Propynoyl Betulin Derivatives against Human Ovarian Cancer Cells: In Vitro Studies"

_ijms, 2023, doi:10.3390/ijms242216487_

Round 1

Reviewer 1 Report (Previous Reviewer 4)

Comments and Suggestions for Authors

The manuscript has evolved considerably since its initial presentation.

However, it bothers me greatly to see that the cell proliferation data (Fig 2 and Fig 3) and the IC50 data (Tab 2) have not had a single statistical analysis carried out. The figures should be analysed by ANOVA and the table by a Tukey test.

Improve the quality of the cell morphology figures.

Comments on the Quality of English Language

Good english language

Author Response

We also appreciate all of the insightful reviewers’ comments and concerns.

  1. However, it bothers me greatly to see that the cell proliferation data (Fig 2 and Fig 3) and the IC50 data (Tab 2) have not had a single statistical analysis carried out. The figures should be analysed by ANOVA and the table by a Tukey test.

Answer:

The quantitative results of SK-OV-3 and OVCAR-3 cells proliferation curves after exposition to tested compounds were analyzed using a non-parametric Kruskal-Wallis test with multiple comparisons. Post-hoc analysis was performed with multiple comparisons of mean ranks test. A p value of <0.05 was considered statistically significant. Detailed results of the statistical analysis were presented in tables as supplementary material. IC50 calculations were performed using the GraphPad Prism software that allows also statistical analyses: the assessment of confidence intervals (95% CI) and determination coefficient – R2. If detailed results are needed, they will be made available at the reviewer's request.

  1. Improve the quality of the cell morphology figures.

Answer:

According to the Reviewer suggestion, the quality of cell morphology figures have been improved.

Reviewer 2 Report (New Reviewer)

Comments and Suggestions for Authors

My comments

Title: Anticancer activity of selected betulin derivatives on human ovarian cancer cell lines

In this manuscript, the author investigated the cytotoxic effect of betulin and its five derivatives against ovarian cancer cells (SK-OV-3, OVCAR-3) and normal myofibroblasts (18Co). They reported that chemical modifications of betulin have greater cytotoxic activity than parent compounds that could be used to develop novel therapeutic systems effective in the treatment of ovarian cancer. I need authors to reply to these comments to get final acceptance for their valuable data.

  1. The introduction must be clear and clear to the study's aim. The less is more. So, the author should discuss the originality of the manuscript.
  2. The author should provide all the Chemical structures of betulin and its five derivatives in the figure and remove the table1.
  3. The author mentioned, "IC50 values (Table 1) were calculated using GraphPad Prism 10.0.2 ", But the author should provide how the IC50 value was calculated for Table 2. 
  4. The author should interpret the Cell proliferation – Sulforhodamine B (SRB) assay clearly. It confuses the reader and provides it in percentage proliferation.
  5. The author should calculate the percentage of apoptotic cells and include it in Figure 6.
  6. The author should provide a quantitative analysis for Figure 7
  7. Why has the author not tested in vivo models? 
Comments on the Quality of English Language

Extensive editing of the English language is required.

Author Response

We also appreciate all of the insightful reviewers’ comments and concerns.

We hope that the revised version of this paper will be accepted for publication in International Journal of Molecular Sciences. If the manuscript is accepted, we will be interested in using a paid editing service to ensure the correctness of the English language used in the manuscript (https://www.mdpi.com/authors/english).

  1. The introduction must be clear and clear to the study’s aim. The less is more. So, the author should discuss the originality of the manuscript.

Answer:

According to the Reviewer suggestion, the introduction has been modified.

  1. The author should provide all the chemical structures of betulin and its five derivatives in the figure and remove the table 1.

Answer:

 According to the Reviewer suggestion, the chemical structures of betulin and its derivatives were placed in Figure 1.

  1. The author mentioned, “IC50 values (Table 1) were calculated using GraphPad Prism 10.0.2, But the author should provide how the IC50 value was calculated for Table 2.

Answer:

IC50 was determined by using “nonlinear regression” mode and "log(inhibitor) vs. normalised response – Variable slope" equation was chosen in GraphPad Prism software for calculations. Obtained data was normalized to run from 100% down to 0% and logarithms of concentration were used. All of the IC50 values were presented in Table 1.

  1. The author should interpret the cell proliferation – Sulforhodamine B (SRB) assay clearly. It confuses the reader and provides it in percentage proliferation.

Answer:

We agree with this reviewer's suggestion, but the aim of this study was to use an in vitro model that would allow observing changes in the proliferation under the influence of incubation with propynoyl derivatives of betulin in a wide range of concentrations and in a long time period – up to 5 days. These results are presented as the proliferation curves. Our idea was to show a comprehensive presentation of cell proliferation during incubation time with the tested compounds in a clear and transparent way. It was important to us to look at the results holistically in order to draw the correct conclusions. The quantitative results of SK-OV-3 and OVCAR-3 cells proliferation curves after exposition to tested compounds were additionally enriched by statistical analysis. Detailed results of the statistical analysis were presented in tables as supplementary material.

  1. The author should calculate the percentage of apoptotic cells and include it in Figure 6.

Answer:

The calculated percentages of apoptotic cells are: 1.3% (control cells); 15.7% (betulin 1); 2.1% (compound 2); 3.3% (compound 3); 3.8% (compound 5) and 1.1% (compound 6).  

  1. The author should provide a quantitative analysis for Figure 7.

Answer:

The calculated percentages of disappearance of red fluorescence (% of TMRM) are: 0% (control); 7.14% (betulin 1); 86.79% (compound 2); 72.51% (compound 3); 74.24% (compound 5) and 79.12% (compound 6).

  1. Why has the author not tested in vivo models?

Answer:

Selected results presented in the manuscript are part of an ongoing project involving the production of modern carriers using the electrospinning technique, and in the next stages of this project their usefulness will be assessed using in vivo models.

Reviewer 3 Report (New Reviewer)

Comments and Suggestions for Authors

The study focuses on finding new therapeutic solutions for ovarian cancer due to its high incidence and limitations of current treatments. The authors evaluated the cytotoxic effect of betulin and its five derivatives against ovarian cancer cells. They found that compounds 2-6 exhibited stronger antiproliferative and cytotoxic activities compared to betulin. Among the tested compounds, compound 2 was the most potent in both ovarian cancer cell lines. It caused a decrease in mitochondrial membrane potential and significant changes in cell morphology. The results suggest that chemical modifications of betulin can lead to compounds with greater cytotoxic activity, which could be used to develop novel therapeutic systems for ovarian cancer treatment. Specific comments:

1.          The title is clear and concise, but it could be more specific about the type of betulin derivatives and the mechanism of their anticancer activity.

2.          The abstract provides a good overview of the background, aim, methods, results and implications of the study. However, it could be improved by:

-          Providing some information about the chemical structure and properties of betulin and its derivatives.

-          Stating the main conclusion more clearly and explicitly.

3.          The introduction gives a comprehensive and relevant background on ovarian cancer, its current treatment options, and the potential of betulin and its derivatives as anticancer agents. However, it could be improved by:

-          Providing more details and references about the molecular mechanisms and pathways involved in ovarian cancer development and resistance to therapy.

-          Explaining the rationale and hypothesis for choosing betulin and its specific derivatives for this study.

4.          The materials and methods section describes the experimental procedures in a clear and reproducible manner. However, it could be improved by:

-          Providing more information about the source, purity, and characterization of betulin and its derivatives.

-          Describing the criteria and methods for selecting the ovarian cancer cell lines and the normal myofibroblasts for this study.

5.          The results section presents the data in a logical and coherent way, using tables, figures, and text. However, it could be improved by:

-          Providing more details and explanations for some of the results, such as the differences in IC50 values between cell lines, the effect of CsA on compound 6-induced cell death, etc.

-          Comparing and contrasting the results with previous studies on betulin and its derivatives in other cancer models or systems.

-          Please provide the scale bars for Figures 6 and 7.

6.          The discussion section interprets the results in light of the existing literature and provides some insights into the molecular mechanisms and implications of betulin derivatives’ anticancer activity. However, it could be improved by:

-          Addressing the limitations and challenges of this study, such as the in vitro nature of the experiments, the lack of mechanistic studies on betulin derivatives’ mode of action, etc.

-          Suggesting some directions and recommendations for future research, such as testing betulin derivatives in vivo models of ovarian cancer, exploring their synergistic effects with other anticancer agents, identifying their molecular targets and pathways, etc.

Author Response

We also appreciate all of the insightful reviewers’ comments and concerns.

We hope that the revised version of this paper will be accepted for publication in International Journal of Molecular Sciences. If the manuscript is accepted, we will be interested in using a paid editing service to ensure the correctness of the English language used in the manuscript (https://www.mdpi.com/authors/english).

  1. The title is clear and concise, but it could be more specific about the type of betulin derivatives and the mechanism of their anticancer activity.

Answer:

According to the Reviewer suggestion, the title has been changed.

  1. The abstract – providing some information about the chemical structure and properties of betulin and its derivatives and stating the main conclusion more clearly and explicitly.

Answer:

According to the Reviewer suggestion, the abstract has been modified.

  1. The introduction – providing more details and references about the molecular mechanisms and pathways involved in ovarian cancer development and resistance to therapy and explaining the rationale and hypothesis for choosing betulin and its specific derivatives for the study.

Answer:

According to the Reviewer suggestion, the introduction has been filled with information about the molecular mechanisms and pathways involved in ovarian cancer development and resistance to therapy [Mohammad, R.M.; Muqbil, I.; Lowe, L.; Yedjou, C.; Hsu, H.Y.; Lin, L.T.; Siegelin, M.D.; Fimognari, C.; Kumar, N.B.; Dou, Q.P., et al. Broad targeting of resistance to apoptosis in cancer. Semin Cancer Biol 2015, 35 Suppl, S78-S103, doi:10.1016/j.semcancer.2015.03.001; Al-Alem, L.F.; Baker, A.T.; Pandya, U.M.; Eisenhauer, E.L.; Rueda, B.R. Understanding and Targeting Apoptotic Pathways in Ovarian Cancer. Cancers (Basel) 2019, 11, doi:10.3390/cancers11111631; Seborova, K.; Vaclavikova, R.; Rob, L.; Soucek, P.; Vodicka, P. Non-Coding RNAs as Biomarkers of Tumor Progression and Metastatic Spread in Epithelial Ovarian Cancer. Cancers 2021, 13, 1839, doi: 10.3390/cancers13081839; Alatise K.L.; Gardner S.; Alexander-Bryant A. Mechanisms of Drug Resistance in Ovarian Cancer and Associated Gene Targets. Cancers (Basel) 2022, 14, 6246, doi: 10.3390/cancers14246246].

  1. The materials and methods – providing more information about the source, purity, and characterization of betulin and its derivatives; and describing the criteria and methods for selecting the ovarian cancer cell lines and the normal myofibroblasts for this study.

Answer:

According to the Reviewer suggestion, more information about betulin, its derivatives and used cell lines have been filled. The presented in the manuscript results are part of broader research on the usefulness of nanocarriers for anticancer drug delivery loaded with acetylenic derivatives of betulin previously tested on HER2-overexpressing SK-BR-3 breast cancer cell line.

  1. The results – providing more details and explanations for some of the results, such as the differences in IC50 values between cell lines, the effect of CsA on compound 6-induced cell death, etc. and comparing and contrasting the results with previous studies on betulin and its derivatives in other cancer models or systems. Please provide the scale bars for Figure 6 and 7.

Answer:

According to literature data [Alatise K.L.; Gardner S.; Alexander-Bryant A. Mechanisms of Drug Resistance in Ovarian Cancer and Associated Gene Targets. Cancers (Basel) 2022, 14, 6246, doi: 10.3390/cancers14246246; Sarwar M.; Sykes P.H.; Chitcholtan K.; Evans J.J. Extracellular biophysical environment: Guilty of being a modulator of drug sensitivity in ovarian cancer cells. Biochem Biophys Res Commun 2020, 527, 180-186, doi: 10.1016/j.bbrc.2020.04.107], the sensitivity to chemotherapy in the SKOV-3 and OVCAR-3 lines may be varied that may confirm the visible differences in the presented results. According to comparative possibilities, the results are discussed with previous results for betulin and its derivatives. The scale bars have been added.

  1. The discussion – addressing the limitations and challenges of this study, such as the in vitro nature of the experiments, the lack of mechanistic studies on betulin derivatives mode of action, etc. and suggesting some directions and recommendations for future research, such as testing betulin derivatives in vivo models of ovarian cancer, exploring their synergistic effects with other anticancer agents, identifying their molecular targets and pathways, etc.

Answer:

Selected results presented in the manuscript are part of an ongoing project involving the production of modern carriers using the electrospinning technique, and in the next stages of this project their usefulness will be assessed using in vivo models.

Round 2

Reviewer 2 Report (New Reviewer)

Comments and Suggestions for Authors

The authors have satisfactorily responded to all comments and made the necessary changes to the manuscript. 

Reviewer 3 Report (New Reviewer)

Comments and Suggestions for Authors

The study focuses on finding new therapeutic solutions for ovarian cancer due to its high incidence and limitations of current treatments. The authors evaluated the cytotoxic effect of betulin and its five derivatives against ovarian cancer cells. They found that compounds 2-6 exhibited stronger antiproliferative and cytotoxic activities compared to betulin. Among the tested compounds, compound 2 was the most potent in both ovarian cancer cell lines. It caused a decrease in mitochondrial membrane potential and significant changes in cell morphology. The results suggest that chemical modifications of betulin can lead to compounds with greater cytotoxic activity, which could be used to develop novel therapeutic systems for ovarian cancer treatment. The revision of the manuscript is improved, no additional comments.

This manuscript is a resubmission of an earlier submission. The following is a list of the peer review reports and author responses from that submission.

Round 1

Reviewer 1 Report

Comments and Suggestions for Authors

Chodurek at al investigated the role of betulin derivatives on human ovarian cancer cell lines. The paper is well-written and provides good results. I would like to recommend it for publication after a minor revision.

- Improve the abstract section by adding results and challenges.

- Mentioning recent publications in the field and highlighting the novelties.

- provide all steps and methods, it seems some steps are not mentioned.

- rewrite the conclusion in a better shape and mention the results and limitation

Reviewer 2 Report

Comments and Suggestions for Authors

In the manuscript “Anticancer activity of selected betulin derivatives on human ovarian cancer cell lines” the authors assessed the in vitro anticancer properties of betulin and of five derivatives with alkyne, phosphate, and phosphonate groups, against two ovarian cancer cell lines, SK-OV-3 and OVCAR-3. The derivatives showed improved activity then betulin, having stronger antiproliferative and cytotoxic effects and inducing morphological changes indicative of regulated cell death stages. The most favorable antitumor properties were reported for C-28 propynoyl derivatives with or without a C-29 diethoxyphosphoryl moiety. The results reported by the authors are of interest for the further development of novel drugs against ovarian cancer. I would recommend manuscript publication after addressing the following issues.

Major issues:

1.     The toxicity of the compounds should be evaluated, and the data should be reported and discussed in the manuscript. The evaluation of the compound safety profile is of key importance for the further compound development into drugs.

2.     The authors should explain why compound 2 has the strongest antiproliferative activity against both cells lines, as assessed using the sulforhodamine B staining (lines 203-205); on the other hand, the most cytotoxic compound was reported to be compound 6 (lines 261-262). Furthermore, there is a low correlation between compound concentrations and cytotoxic effects in the cytotoxicity data displayed in Figure 3. Unusual trends are also observed in the data displayed in Figure 4 The authors should explain this in the manuscript.

3.     Measurement should be always displayed reporting errors (e.g., Table 1) and error bars (e.g., Figures 1 and 2)

Minor issues:

1.     Abstract, line 25. The abbreviation RCD should be explained.

Page 1, lines 34-35. Correct the number format “313, 959” in “313,959” and “207, 252” in “207,252”.

2.     Page 2, lines 66-68. The sentence is not clear, please rephrase. 

3.     Section 2, lines 122-133. This part has been already discussed in the introduction, especially the first sentence that is part of the introduction, lines 31-35.  

4.     Figure 8 and Table 2, showing the chemical structures of the studied compounds should be displayed at the beginning of section 2 (referred to lines 143-144).

5.     Page 4, line 162. Change “that the compound 2” to “that compound 2”.

6.     Error bars should be displayed in Figures 1 and 2.

7.     Page 6, line 221. “2+” should be superscript.

8.     Page 6, lines 222-224. The sentence is not clear, please rephrase. 

9.     Page 9, line 297. Change “loss a part” to “loss of a part”.

Reviewer 3 Report

Comments and Suggestions for Authors

The paper presents the antiproliferative effect of betulin and five of its derivatives on two human ovarian cancer cell lines (SK-OV-3 and OVCAR-3). The compounds are known and some of them have previously shown potent antiproliferative activity against various tumor cell lines.

The discovery is not attractive enough for publication in its current version in such a reputable journal as IJMS. The article requires corrections:

Introduction: The authors did not emphasize the novelty and importance of the research conducted

Additional studies are necessary :

a) the cytotoxicity of betulin derivatives against normal non-cancerous cells should be assessed to exclude their general toxicity

b) the antiproliferative activity of betulin derivatives should be compared with an FDA-approved drug used in ovarian cancer therapy

Results and discussion: lines 122-142 - these paragraphs contain information that is already in the Introduction section or should be moved into the Introduction section

Structure-activity relationships should be accurately described

Conclusions: this section needs to be modified as it does not emphasize the importance of the research carried out, and even reduces its value

The Abstract should be modified, because only the second part of it presents the research carried out

Abstract: the RCD abbreviation should be explained

Reviewer 4 Report

Comments and Suggestions for Authors

Do not use abbreviations in the abstract;

The introduction is excessively long. The text can be condensed, making it more objective and direct;

I recommend that the results section be separated from the discussion, as recommended by the author guidelines;

I strongly recommend using statistical software in the analysis of the cell proliferation curve (Figures 1 and 2) for both data presentation and statistical analysis. The authors aim for publication in a journal with an IF of 6.20 and present Excel graphs? These figures are not from statistical software.

Please also present the deviations of these readings.

Table 1 - Where are the deviations, confidence intervals, and R²? And statistical analysis of this data?

Figure 3 - the same recommendation as for Figures 1 and 2. * indicates comparison with control. Which control? Positive or negative? Why is it not shown in the figure? It MUST be.

I am very concerned about the use of only three concentrations; it is difficult for an IC50 calculation and dose-dependency assessment to be reliable under this condition.

Figures 5 and 6 - the images are of poor quality and having only 100X magnification is concerning for obtaining the data that the authors describe. There is a lack of an adequate description of how these cells were counted.

Figure 7 - the same as for Figures 1 and 2.

This morphological analysis study alone (which was very superficial) is insufficient to at least suggest that these derivatives can be considered for further stages.

Assays evaluating the effects of these derivatives on Cell Cycle, DNA Content Analysis by Flow Cytometry, and assessment of the apoptosis mechanism by Western Blotting are mandatory for this type of study.